# Readability, quality, and reliability of AI-generated ınformation on myofascial pain syndrome: A comparative analysis of ChatGPT, Gemini, and Perplexity

Yüksel Erkin[1], Erkan Ozduran[2]*, İlhan Celil Özbek[3], Volkan Hancı[4]

**1** Dokuz Eylul University, Anesthesiology and Reanimation, Pain Medicine, Izmir, Turkey, **2** Sivas Numune Hospital, Physical Medicine and Rehabilitation, Pain Medicine, Sivas, Turkey, **3** Health Science University, Derince Training and Research Hospital, Physical Medicine and Rehabilitation, Kocaeli, Turkey, **4** Dokuz Eylul University, Anesthesiology and Reanimation, Critical Care Medicine, Izmir, Turkey

* erkanozduran@gmail.com

## Abstract

Patients seeking information about Myofascial Pain Syndrome (MPS), which affects a large segment of the population, are increasingly turning to AI-based chatbots as an alternative to traditional methods. However, the medical accuracy of the content offered by these digital platforms, as well as its suitability to the "grade 6 reading level" standard, which determines its comprehensibility by patients, is a critical point of uncertainty. This study aims to fill this significant gap in the literature by systematically comparing MPS content generated by different AI models using readability indices, reliability, and quality metrics. The 18 most relevant keywords, derived from 25 keywords identified via Google Trends data, were queried using ChatGPT (GPT-5.2), Gemini 3 Flash, and Perplexity (Sonar-4 Large) models. The readability of the generated responses was analyzed using six different indices (FRES, FKGL, GFOG, CLI, ARI, SMOG), while content quality was assessed using GQS and EQIP scales, and reliability using DISCERN and JAMA scales by two independent observers. The responses generated by all AI models examined were found to be statistically significantly more complex than the suggested 6th-grade reading level (p < 0.001). In inter-model comparisons, ChatGPT exhibited the easiest readability [lowest linguistic difficulty] scores, while Perplexity scored significantly higher than both ChatGPT and Gemini in content quality and reliability metrics (JAMA, DISCERN, GQS, EQIP) (p < 0.05). Correlation analysis revealed a strong and positive relationship between quality and reliability parameters. Artificial intelligence platforms have been observed to exhibit high potential in the production of medical information. However, linguistic barriers exceeding sixth-grade reading comprehension, along with reliability limitations of current models, prevent them from replacing professional medical consultation. Perplexity has been found superior in terms of academic quality, while ChatGPT has been found superior in terms of readability. Nevertheless, positioning these systems as

**Data availability statement:** All relevant data required to replicate the findings of this study are fully available without restriction. The minimal anonymized dataset, including all raw prompt queries and model responses analyzed during the study, has been deposited in the public repository Figshare. It can be freely and directly accessed via the following direct link: https://figshare.com/articles/online_resource/myofascial_pain_syndrome/31583908?-file=62548162. Additionally, the complete dataset (raw data) has been uploaded directly to the journal system as a Supporting information file.

**Funding:** The author(s) received no specific funding for this work.

**Competing interests:** The authors have declared that no competing interests exist.

complementary "secondary consultation mechanisms" supporting physician oversight in clinical decision-making processes is critically important for patient safety.

## Introduction

Myofascial pain syndrome (MPS) is a common musculoskeletal pain disorder characterized by palpable trigger points and characteristic referred pain patterns in the skeletal muscles and associated facial structures [1]. Clinically, regional pain is often accompanied by muscle spasm, limited range of motion, and decreased functional capacity; this can significantly affect patients' daily activities and quality of life [2]. While it can occur in all age groups, it is reported more frequently in individuals exposed to repetitive microtrauma, prolonged static posture, and occupational strain [3]. Epidemiological data show that approximately 85% of the general population experiences myofascial pain at some point in their lives; this high rate suggests that MPS is one of the leading causes of chronic musculoskeletal pain [4]. This prevalence indicates that the syndrome constitutes a significant health burden not only at the individual level but also at the societal level.

In recent years, the use of artificial intelligence (AI) technologies in the healthcare field has increased significantly. With this development, AI-based systems have become more frequently used in the production of digital content and patient information related to chronic pain syndromes. In particular, chatbots have become platforms where individuals try to interpret their symptoms, search for information about possible diagnoses, and explore treatment options [5]. While studies examining the accuracy and reliability of AI-generated content in the fields of orthopedics, rheumatology, and chronic pain are increasing, the extent to which this content is scientifically adequate and suitable for patient safety remains unclear [6–8]. Incorrect or incomplete information presentation can lead to unnecessary anxiety, erroneous self-management attempts, and treatment adherence problems. Therefore, systematic and critical evaluation of AI-derived health information is clinically important.

The accuracy of health-related information is as important as its comprehensibility by the target audience. Readability is assessed through criteria that indicate how well a text aligns with the reader's educational level and language skills. For this purpose, formulas such as Flesch-Kincaid, Gunning Fog, and SMOG are frequently used, and texts are classified according to educational level. The American Medical Association (AMA), the National Institutes of Health ([NIH], and the U.S. Department of Health and Human Services recommend that patient education materials be prepared at a sixth-grade level or lower [9,10]. Given that MPS is a widespread condition affecting a broad segment of the population, it is particularly important that digital health information related to this disease is presented in a way that is understandable to everyone. Considering that individuals from different educational and socioeconomic backgrounds access this content, preparing the information in clear and simple language can be crucial for patients to accurately assess their disease process and treatment options [11,12].

Studies examining health content generated by AI-powered chatbots regarding MPS are quite limited. The current literature indicates a need for comprehensive and comparative evaluations in this area. This study aims to systematically evaluate the quality and readability of texts generated by different AI-powered chatbots regarding MPS. The results are expected to provide a clearer framework for understanding the quality of health information in the digital environment and contribute to the development of patient-centered information approaches in chronic pain management.

## Materials and methods

**Ethical Assessment:** Since the study's methodology did not require application to live subjects, it was evaluated outside the scope of studies requiring ethics committee approval, in accordance with similar scientific standards. Data collection and analysis were performed in accordance with the terms and conditions of the data source and institutional guidelines. The study was conducted in strict adherence to the principles of the Declaration of Helsinki, and all data were processed and analyzed in compliance with the relevant data protection regulations.

### Research design

To prevent potential algorithmic biases and the influence of individual user preferences on the results during the data collection process, browser history and cookies were completely cleared before analysis. The research was conducted without the use of a VPN, via Google's "Incognito Mode" tab, to minimize the impact of historical data. Global interest and geographical distribution data for the keyword "Myofascial Pain Syndrome" were obtained via the Google Trends platform on March 1, 2026. Accordingly, global data from 2004 to the present were scanned using the "most relevant" filter; the 25 most frequently used keywords and their regional concentrations were documented [13]. The reason for choosing Google as the primary data source in this study is that the platform offers the most comprehensive database with a global market share of 82.24% according to October 2025 data [14].

### AI analysis protocol

This research aims to qualitatively analyze the responses provided by three widely accessible AI platforms—ChatGPT, Perplexity, and Gemini—to keywords related to "myofascial pain syndrome" obtained from Google Trends data. The relevant terms were submitted to the systems as English queries [9,15]. To ensure methodological consistency, all models were evaluated using the same query expressions, and the initial response obtained for each keyword was recorded for analysis. To prevent possible systematic biases and the influence of previous queries on the models [contextual deviation], independent and new user sessions were opened for each keyword. The obtained outputs were classified and recorded according to the criteria of content accuracy, reliability, and readability. All raw data and model responses used in the analysis process are accessible through the digital archive determined in accordance with open science principles: (https://figshare.com/articles/online_resource/myofascial_pain_syndrome/31583908?file=62548162). The study was based on the current free version of ChatGPT (GPT-5.2) as of December 9, 2025; free versions of Gemini 3 Flash and Perplexity (Sonar-4 Large) were also included in the comparative analysis to support the universality of the findings [9,16].

In our study, repeated/synonymous words and non-MPS keywords were determined as exclusion criteria, while English keywords related to MPS were determined as inclusion criteria [2,8].

### Readability assessment of texts

To ensure data consistency and reliability in the readability analysis of responses obtained from AI models, cross-checking was performed across two different independent platforms. Accordingly, all texts were simultaneously evaluated using both readabilityformulas.com (Calculator 1) and online-utility.org (Calculator 2) digital tools. The use of two different calculation interfaces and their average aims to minimize potential technical deviations arising from application variations

of the metrics and to confirm the methodological validity of the obtained scores. The evaluation process was based on six different metrics whose reliability has been established in the academic literature: Flesch Readability Score [FRES], Flesch-Kincaid Class Level (FKGL), Gunning Fog Index (GFOG), Coleman-Liau Index (CLI), AutoReadability Index (ARI), and Gobbledygook Simple Criterion (SMOG). The simultaneous use of these formulas has made it possible to comprehensively analyze the suitability of the produced content for the general readership in terms of both word level and sentence structure [8,9,17,18].

The analysis results were reported using median values and minimum-maximum ranges to represent the overall comprehensibility level of the texts. The data obtained were compared with the "sixth-grade reading level" criterion recommended by the American Medical Association (AMA) and the National Institutes of Health (NIH) for public health information. Accordingly, while 80.0 points was accepted as the threshold value for the FRES metric; the sixth-grade [6.0] level was determined as the target achievement criterion for the FKGL, GFOG, CLI, ARI and SMOG indices [9,17].

### Reliability assessment protocol

Two main assessment tools were used to measure the quality and academic consistency of the information provided by AI systems. First, the Modified DISCERN Scale, which tests source reliability in five different dimensions, was applied [18]. Within this scale, parameters such as referencing, timeliness, level of transparency, approach to controversial issues, and objectivity were rated between 0–5 points. An increase in the total score was directly related to the high level of reliability and academic quality of the information provided [19]. The methodological validity of standardized measurement tools such as DISCERN and JAMA [Journal of the American Medical Association] is also supported by previous studies in the field [20,21].

The JAMA questionnaire was used as the second reliability criterion of the research. In this context, each response was checked against four basic publishing ethics principles: authorship, currency, disclosure, and attribution [22]. In the evaluation process based on JAMA criteria, a binary scoring system [0 or 1] was adopted for each criterion; the final score obtained [maximum 4 points] revealed the capacity of the relevant content to comply with academic standards. High scores were considered an indicator of the platform's commitment to ethical publishing principles and scientific reliability norms [6,23].

### Content quality assessment

**Global Quality Scale (GQS) analysis.** To measure the quality of digital health content and the clinical benefit it provides to the user, the Global Quality Score (GQS), which has widespread validity in the literature, has been used [5,24,25]. This evaluation tool classifies materials on a five-point Likert scale ranging from 1 to 5. In the scoring system, a score of 1 represents low-quality and inadequate content that has no value for patients; a score of 5 represents high-quality material that is highly reliable, comprehensive, and provides direct clinical benefit to the patient.

The scoring at the intermediate levels is structured according to the following criteria: 2 points indicate low quality but limited potential for use; 3 points indicate medium quality and limited information; and 4 points indicate high quality and clinically significant benefit [5,24]. The methodological reliability of the GQS instrument and its applicability in various digital health research have also been confirmed by previous scientific studies [26].

Another quality assessment tool used in the study is the EQIP [Ensuring Quality Information for Patients] scale, which allows for the systematic monitoring of the quality of medical texts from a patient perspective. This methodological instrument analyzes the content under review through a comprehensive set of 20 items; each item is rated as "yes", "partially" or "no" with 1, 0.5 and 0 points, respectively. The final quality score is calculated as a percentage value obtained by dividing the total score by the number of applicable items and multiplying the result by 100 [27]. The percentage data resulting from this calculation classifies the qualitative level of the content into four hierarchical categories: 76–100% represents high-quality "well-written" texts; 51–75% represents content of "good quality with minor structural problems"; 26–50%

 

represents materials containing "significant problems"; and 0–25% represents content with "severe problems" in terms of clinical reliability [8].

All reliability and quality assessments were carried out separately by two independent researchers (Y.E. and E.O.) familiar with the relevant scales; consistency between the evaluators was statistically analyzed.

## Statistical analysis

All data generated during the study were processed and analyzed using IBM SPSS Statistics software (Version 24.0; IBM Corp., Armonk, NY, USA). Categorical variables were expressed as frequencies and percentages, whereas continuous variables were reported as medians with their corresponding ranges (minimum–maximum).

Comparisons among the three AI models were performed using the non-parametric Kruskal–Wallis test. When statistically significant differences were detected, pairwise comparisons were conducted using the Mann–Whitney U test with Bonferroni correction to control for multiple comparisons. Additionally, comparisons between the observed readability scores and the recommended sixth-grade readability level (grade 6.0) were performed using the one-sample Wilcoxon signed-rank test.

Correlations between quality and reliability metrics were assessed using Spearman's rank correlation coefficient. For quality and reliability assessments, inter-observer agreement between the two independent evaluators [Y.E. and E.O.], who were also the authors of the study, was evaluated using the Intraclass Correlation Coefficient (ICC).

Correlations between quality and reliability metrics were assessed using Spearman's rank correlation coefficient. For quality and reliability assessments, inter-observer agreement between the two independent evaluators (Y.E. and E.O.), who were also the authors of the study, was evaluated using the Intraclass Correlation Coefficient (ICC) with 95% confidence intervals (CIs). ICC values were interpreted as follows: $<0.50$ poor, 0.50–0.75 moderate, 0.75–0.90 good, and $>0.90$ excellent reliability.

For all statistical analyses, a p-value of less than 0.05 was considered statistically significant. For post hoc pairwise comparisons, Bonferroni correction was applied; given that three pairwise comparisons were performed, the adjusted significance threshold was set at $p<0.0167$. Reported p-values represent unadjusted values, and statistical significance was interpreted according to the Bonferroni-adjusted threshold.

## Results

Google Trends data was analyzed to determine the trends of users searching for information about MPS on the Google platform. Within the scope of the research design, 18 unique queries were directed to each of three different AI models, resulting in a total of 54 datasets. These generated responses were individually examined in terms of readability, content quality, and reliability parameters, and the performance of each model was calculated based on the median values of these 18 responses. As a result of the analyses, it was determined that the most frequently used basic keywords were "muscle pain", "icd 10 myofascial pain syndrome", and "myofascial pain syndrome symptoms". The basic terms that constitute the data pool of the study are: "muscle pain", "icd 10 myofascial pain syndrome", "myofascial pain syndrome symptoms", "myofascial pain syndrome treatment", "chronic myofascial pain", "myofascial release", "chronic myofascial pain syndrome", "cervical myofascial pain syndrome", "myofascial pain dysfunction syndrome", "what is myofascial pain syndrome", "myofascial trigger points", "myofascial pain syndrome trigger points", "myofascial pain syndrome causes", "fibromyalgia and myofascial pain syndrome", "myofascial MPSsage", "myofascial pain syndrome diagnosis", "fascia pain" and "myofascial pain syndrome disability". However; The expressions "fibromyalgia", "myofascial pain syndrome icd10", "myofascial pain symptoms", "myofascial pain treatment", "chronic pain syndrome" and "neck pain" were excluded from the analysis process. This exclusion criterion is based on methodological grounds such as some terms being semantically repetitive, not directly aligning with the scope of the study, or AI systems producing off-focus responses to these queries.

Eighteen keywords were evaluated in the study. Analyses conducted using Google Trends data revealed that the highest search volume for the term "myofascial pain syndrome" originated from the Philippines, Thailand, and the United States, respectively. Global geographic interest level and regional distribution data for this keyword are visualized in Fig 1. These identified MPS-related keywords were submitted as query input to the Perplexity, ChatGPT, and Google Gemini platforms.

The readability performance of the response texts obtained from these AI systems was calculated using Calculator 1, Calculator 2, and the arithmetic mean of these two measurements. The obtained readability levels were subjected to a comparative analysis based on the "grade 6 reader level" standard, which is accepted as a reference for medical informational materials by the American Medical Association Foundation and the American Medical Association (Tables 1–3).

### Readability analysis findings based on calculator 1 data

During the text readability analysis conducted using Calculator 1 for 18 different responses to 18 questions, a statistically significant difference was found between the median readability values of all responses and the sixth-grade readability across all three AI chatbots ($p < 0.001$). Significant differences were found between ChatGPT and Gemini in the FRES ($p < 0.001$), GFOG ($p < 0.001$), and CLI ($p < 0.001$) formulas; between ChatGPT and Perplexity in the GFOG ($p = 0.015$) and ARI ($p = 0.005$) formulas; and between Gemini and Perplexity in the GFOG ($p = 0.015$) and ARI ($p = 0.005$) formulas (Table 1).

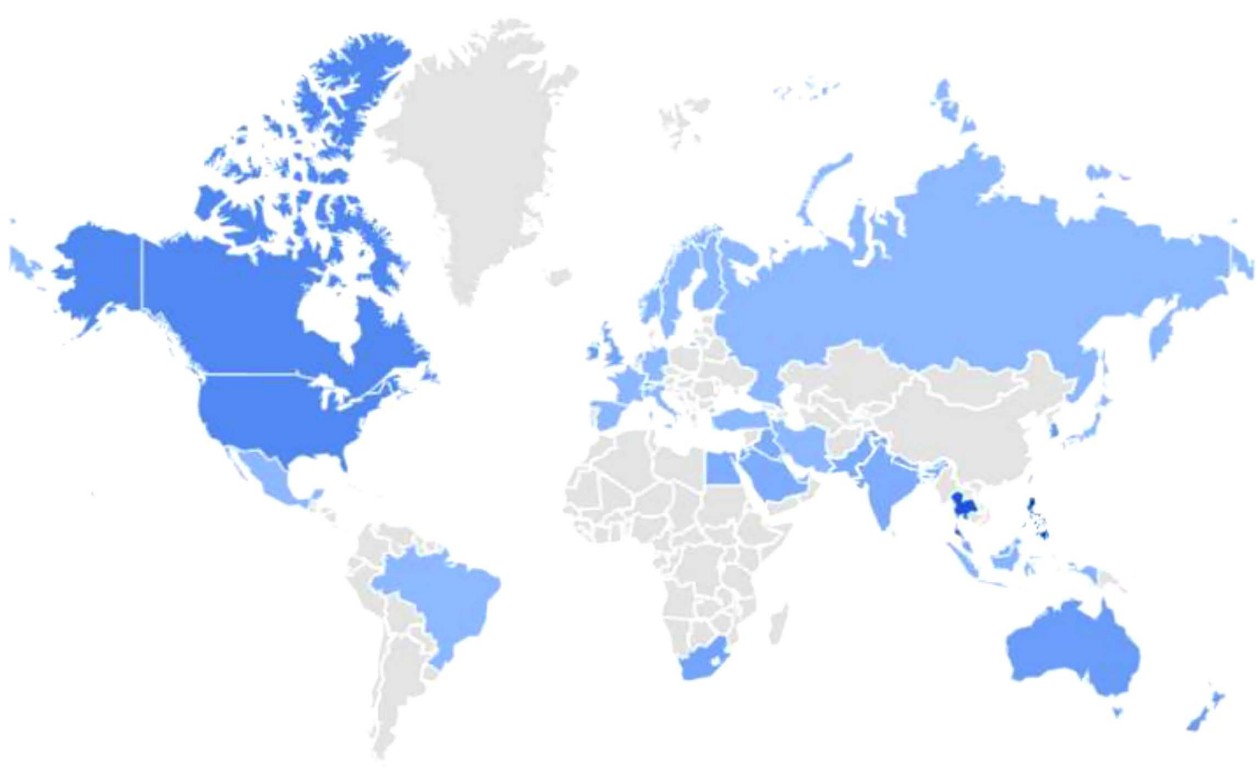

**Fig 1. Search interest in Myofascial Pain Syndrome across the countries: 2004-present (Based on Google Trends Data, https://www.google.com/trends).**

**Table 1. Readability Scores (Calculator-1).**

| CALCULATOR Statistics | ChatGPT | Gemini | Perplexity | ChatGPT C6thGRL (P)** † | Gemini C6thGRL (P)*- † | Perplexity C6thGRL (p)** † | Between ChatGPT and Gemini (p)†† | Between ChatGPT and Perplexity (p)†† | Between Perplexity and Gemini (p)†† |
|---|---|---|---|---|---|---|---|---|---|
| FRES | 34.61 ± 11.41 (13.00–57.00) | 53.28 ± 9.55 (32.00–75.00) | 37.89 ± 8.07 (23.00–56.00) | **<.001** | **<.001** | **<.001** | **<0.001** | 0.281 | 0.281 |
| GFOG | 13.02 ± 1.66 (8.80–15.30) | 10.38 ± 1.47 (7.90–13.80) | 11.64 ± 1.29 (8.20–13.80) | **<.001** | **<.001** | **<.001** | **<0.001** | **0.005** | **0.005** |
| FKGL | 10.77 ± 1.50 (7.91–13.69) | 9.64 ± 1.43 (6.33–12.13) | 11.30 ± 1.32 (8.06–13.22) | **<.001** | **<.001** | **<.001** | 0.037 | 0.150 | 0.150 |
| CLI | 15.31 ± 2.06 (10.82–18.87) | 12.25 ± 1.88 (7.15–15.53) | 16.03 ± 1.62 (12.01–18.66) | **<.001** | **<.001** | **<.001** | **<0.001** | 0.229 | 0.229 |
| SMOG | 8.96 ± 1.10 (6.88–11.57) | 8.85 ± 1.07 (6.97–11.01) | 9.57 ± 0.98 (7.01–11.01) | **<.001** | **<.001** | **<.001** | 0.950 | 0.066 | 0.066 |
| ARI | 12.25 ± 1.57 (9.16–15.07) | 11.32 ± 1.54 (7.11–13.70) | 13.55 ± 1.56 (10.24–16.19) | **<.001** | **<.001** | **<.001** | 0.100 | **0.015** | **0.015** |

Calculator 1: https://readabilityformulas.com/free-readability-formula-tests.php.

Calculator 2: https://www.online-utility.org/english/readability_test_and_improve.jsp.

Abbreviations: Flesch reading ease score (FRES), Gunning FOG (GFOG), Flesch-Kincaid Grade Level (FKGL), Simple Measure of Gobbledygook (SMOG), Coleman-Liau Index (CLI), Automated Readability Index (ARI) and Linsear Write (LW).

**: C6thGRL**(p):** Comparison of the responses according to 6th grade reading level **(p).**

†: Wilcoxon test.

††: Chi-Square test for categorical variables and Mann-Whitney U test for continuous variables.

p *values* in *bold* are *statistically significant*.

Pairwise comparisons were performed using the Mann–Whitney U test with Bonferroni correction. Reported p-values are unadjusted; statistical significance was interpreted using a Bonferroni-adjusted threshold of p<0.0167. Bold p-values indicate statistically significant results.

## Readability analysis findings based on calculator 2 data

During the text readability analysis using calculator 2 for 18 different answers to 18 questions, a statistically significant difference was found between the median readability values of all answers and the sixth-grade readability across all three AI chatbots (p<0.001). Significant differences were detected between ChatGPT and Gemini in the FRES formula (p=0.001), between ChatGPT and Perplexity in the FKGL formulas (p=0.005), CLI formulas (p=0.001), SMOG formulas (p<0.001), and ARI formulas (p<0.001), and between Gemini and Perplexity in the FRES formulas (p<0.001), FKGL formulas (p=0.002), CLI formulas (p<0.001), SMOG formulas (p=0.005), and ARI formulas (p=0.001) (Table 2).

## Text readability analysis based on the average of calculator 1 and 2

During the text readability analysis using the averages of calculators 1 and 2 for 18 different answers to 18 questions, a statistically significant difference was found between the median readability values of all answers and sixth-grade readability across all three AI chatbots (p<0.001). Significant differences were detected between ChatGPT and Gemini in the FRES (p<0.001), GFOG (p=0.005), and CLI (p=0.003) formulas; between Gemini and Perplexity in the FRES (p<0.001), GFOG (p=0.009), FKGL (p=0.001), CLI (p<0.001), SMOG (p=0.006), and ARI (p<0.001) formulas; and between ChatGPT and Perplexity in the CLI (p=0.015), SMOG (p<0.001), and ARI (p<0.001) formulas.

Interpretation of the statistical data revealed that ChatGPT-5 generated responses with the lowest linguistic complexity and highest accessibility; other models were hierarchically ranked as Gemini and Perplexity (Table 3).

**Table 2. Readability Scores (Calculator-2).**

| CALCULATOR Statistics | ChatGPT | Gemini | Perplexity | ChatGPT C6thGRL (P)** † | Gemini C6thGRL (P)*- † | Perplexity C6thGRL (p)** † | Between ChatGPT and Gemini (p)†† | Between ChatGPT and Perplexity (p)†† | Between Perplexity and Gemini (p)†† |
|---|---|---|---|---|---|---|---|---|---|
| FRES | 34.82 ± 11.36 (13.66–58.27) | 48.88 ± 8.92 (26.55–65.21) | 32.56 ± 8.76 (19.78–47.30) | <.001 | <.001 | <.001 | **0.001** | 0.527 | **<0.001** |
| GFOG | 13.00 ± 1.92 (8.90–17.01) | 12.50 ± 1.52 (10.77–15.72) | 13.99 ± 1.89 (11.12–17.17) | <.001 | <.001 | <.001 | 0.289 | 0.179 | 0.019 |
| FKGL | 10.87 ± 1.87 (7.66–15.35) | 10.66 ± 1.50 (8.06–13.64) | 12.79 ± 1.88 (8.90–15.67) | <.001 | <.001 | <.001 | 0.752 | **0.005** | **0.002** |
| CLI | 13.54 ± 2.50 (8.18–18.12) | 12.10 ± 1.92 (7.89–15.74) | 16.31 ± 2.00 (11.84–19.25) | <.001 | <.001 | <.001 | 0.052 | **0.001** | **<0.001** |
| SMOG | 11.94 ± 1.46 (9.20–15.87) | 12.36 ± 1.21 (10.39–14.42) | 13.81 ± 1.40 (11.32–16.19) | <.001 | <.001 | <.001 | 0.255 | **0.001** | **0.005** |
| ARI | 9.80 ± 2.37 (5.19–15.10) | 10.81 ± 2.02 (6.11–13.49) | 13.96 ± 2.64 (8.67–18.14) | <.001 | <.001 | <.001 | 0.076 | **<0.001** | **0.001** |

Calculator 1: https://readabilityformulas.com/free-readability-formula-tests.php.

Calculator 2: https://www.online-utility.org/english/readability_test_and_improve.jsp.

Abbreviations: Flesch reading ease score (FRES), Gunning FOG (GFOG), Flesch-Kincaid Grade Level (FKGL), Simple Measure of Gobbledygook (SMOG), Coleman-Liau Index (CLI), Automated Readability Index (ARI) and Linsear Write (LW).

**: C6thGRL**(p):** Comparison of the responses according to 6th grade reading level **(p).**

†: Wilcoxon test.

††: Chi-Square test for categorical variables and Mann-Whitney U test for continuous variables.

p *values* in *bold* are *statistically significant*.

Pairwise comparisons were performed using the Mann–Whitney U test with Bonferroni correction. Reported p-values are unadjusted; statistical significance was interpreted using a Bonferroni-adjusted threshold of p<0.0167. Bold p-values indicate statistically significant results.

**Content Quality and Source Reliability Analysis Findings** A detailed breakdown of the GQS, EQIP, DISCERN, and JAMA scores of the responses generated by AI-based chatbots is presented in Table 4. In the overall evaluation, which included three different AI models, statistically significant differences were found in all JAMA, DISCERN, GQS, and EQIP parameters. In pairwise comparison analyses, Perplexity achieved significantly higher scores in the GQS (p = 0.001), JAMA (p < 0.001), mDISCERN (p = 0.001), and EQIP (p = 0.001) scales when compared to ChatGPT-5. Similarly, Perplexity also showed statistically better performance in the GQS (p = 0.001), JAMA (p < 0.001), mDISCERN (p < 0.001), and EQIP (p < 0.001) questionnaires according to the Gemini model. In the comparison between Gemini and ChatGPT, a significant difference was found only in the JAMA scoring (p < 0.001). These analyses demonstrated that Perplexity scored significantly higher than both ChatGPT and Gemini in all evaluation tools. Furthermore, Gemini scored higher than ChatGPT specifically in the JAMA criteria. In conclusion, Perplexity offers the highest level of quality and reliability; in the comparison between Gemini and ChatGPT, reliability favored Gemini only based on the JAMA results.

Inter-rater agreement was high in quality and reliability assessments. Intraclass Correlation Coefficient values were as follows: mDISCERN 0.761, JAMA 0.876, GQS 0.795, and EQIP 0.999, indicating good to excellent reliability according to predefined ICC interpretation thresholds.

## Correlation analysis

The correlation analysis performed revealed a statistically strong and positive relationship between the survey metrics measuring reliability and content quality. This finding indicates that the quality (EQIP, GQS) and reliability (mDISCERN,

**Table 3. Readability Scores (Average of Calculators).**

| CALCULATOR Statistics | ChatGPT | Gemini | Perplexity | ChatGPT C6thGRL (P)** † | Gemini C6thGRL (P)*- † | Perplexity C6thGRL (p)** † | Between ChatGPT and Gemini (p)†† | Between ChatGPT and Perplexity (p)†† | Between Perplexity and Gemini (p)†† |
|---|---|---|---|---|---|---|---|---|---|
| FRES | 34.72 ± 11.25 (13.33–57.64) | 51.08 ± 9.11 (29.28–70.10) | 35.22 ± 8.08 (21.75–50.67) | <.001 | <.001 | <.001 | <0.001 | 0.776 | <0.001 |
| GFOG | 13.01 ± 1.73 (8.85–15.91) | 11.44 ± 1.41 (9.39–14.63) | 12.82 ± 1.45 (9.77–15.33) | <.001 | <.001 | <.001 | 0.005 | 0.658 | 0.009 |
| FKGL | 10.82 ± 1.64 (8.03–14.52) | 10.15 ± 1.39 (7.28–12.89) | 12.04 ± 1.46 (9.35–14.34) | <.001 | <.001 | <.001 | 0.217 | 0.018 | 0.001 |
| CLI | 14.43 ± 2.22 (9.50–18.50) | 12.18 ± 1.88 (7.52–15.64) | 16.17 ± 1.77 (11.93–18.96) | <.001 | <.001 | <.001 | 0.003 | 0.015 | <0.001 |
| SMOG | 10.45 ± 1.13 (8.44–13.72) | 10.60 ± 1.07 (8.68–12.72) | 11.69 ± 1.03 (9.64–13.47) | <.001 | <.001 | <.001 | 0.752 | <0.001 | 0.006 |
| ARI | 11.02 ± 1.87 (7.92–15.09) | 11.07 ± 1.67 (7.14–13.56) | 13.75 ± 1.91 (9.82–17.13) | <.001 | <.001 | <.001 | 0.681 | <0.001 | <0.001 |

Calculator 1: https://readabilityformulas.com/free-readability-formula-tests.php.

Calculator 2: https://www.online-utility.org/english/readability_test_and_improve.jsp.

Abbreviations: Flesch reading ease score (FRES), Gunning FOG (GFOG), Flesch-Kincaid Grade Level (FKGL), Simple Measure of Gobbledygook (SMOG), Coleman-Liau Index (CLI), Automated Readability Index (ARI) and Linsear Write (LW).

**: C6thGRL(p): Comparison of the responses according to 6th grade reading level (p).

†: Wilcoxon test.

††: Chi-Square test for categorical variables and Mann-Whitney U test for continuous variables.

p *values* in *bold* are *statistically significant*.

Pairwise comparisons were performed using the Mann–Whitney U test with Bonferroni correction. Reported p-values are unadjusted; statistical significance was interpreted using a Bonferroni-adjusted threshold of p < 0.0167. Bold p-values indicate statistically significant results.

JAMA) parameters methodologically support each other. In addition, significant correlation coefficients were found between different readability indices; this indicates that text complexity yields consistent results in measurements made with different formulas (Table 5). Detailed data on the strengths and significance levels of these correlations are presented in the relevant table.

## Discussion

This research reveals that the responses of leading AI-based language models (Perplexity, Gemini, and ChatGPT) to questions about MPS have a complexity exceeding the "6th-grade reading level" standard recommended by the U.S. Department of Health and Human Services (HHS) and the National Institutes of Health (NIH). Data from the GQS, EQIP, DISCERN, and JAMA scales demonstrate that ChatGPT exhibits the most optimized performance in terms of readability, followed by Gemini and Perplexity, respectively. While Perplexity achieved the highest scores in content quality and reliability parameters compared to the other models, a comparison between Gemini and ChatGPT showed that reliability was only in favor of Gemini based on JAMA criteria, with no significant difference in quality scores. The MPS content generated by these AI platforms was subjected to a comprehensive evaluation using DISCERN/JAMA tools for reliability and GQS/EQIP tools for quality and cognitive accessibility (readability). Our current study represents a unique and strategic contribution to the contemporary scientific literature as it is one of the pioneering studies that comparatively analyzes the responses of the three most popular major language models (LLMs) regarding MPS.

**Table 4. Comparison Of Quality and Reliability Ratings for the Responses from AI Chatbots.**

| | ChatGPT vs Perplexity | | | ChatGPT vs Gemini | | | Perplexity vs Gemini | | |
|---|---|---|---|---|---|---|---|---|---|
| | ChatGPT | Perplexity | P | ChatGPT | Gemini | P | Perplexity | Gemini | P |
| **GQS, n (%)** | | | **0.001** | | | .736 | | | **0.001** |
| 1-point | 2 (11,1%) | 0 (0%) | | 2 (11,1%) | 0 (0%) | | 0 (0%) | 0 (0%) | |
| 2-point | 3 (16,7%) | 0 (0%) | | 3 (16,7%) | 4 (22,2%) | | 0 (0%) | 4 (22,2%) | |
| 3-point | 7 (38,9%) | 2 (11,1%) | | 7 (38,9%) | 8 (44,4%) | | 2 (11,1%) | 8 (44,4%) | |
| 4-point | 6 (33,3%) | 16 (88,9%) | | 6 (33,3%) | 6 (33,3%) | | 16 (88,9%) | 6 (33,3%) | |
| 5-point | 0 (0%) | 0 (0%) | | 0 (0%) | 0 (0%) | | 0 (0%) | 0 (0%) | |
| **JAMA, n (%)** | | | **<.001** | | | **<.001*** | | | **<.001** |
| 0-point | 11 (61,1%) | 0 (0%) | | 11 (61,1%) | 3 (16,7%) | | 0 (0%) | 3 (16,7%) | |
| 1-point | 7 (38,9%) | 0 (0%) | | 7 (38,9%) | 9 (50,0%) | | 0 (0%) | 9 (50,0%) | |
| 2-point | 0 (0%) | 8 (44,4%) | | 0 (0%) | 6 (33,3%) | | 8 (44,4%) | 6 (33,3%) | |
| 3-point | 0 (0%) | 10 (55,6%) | | 0 (0%) | 0 (0%) | | 10 (55,6%) | 0 (0%) | |
| 4-point | 0 (0%) | 0 (0%) | | 0 (0%) | 0 (0%) | | 0 (0%) | 0 (0%) | |
| **m DISCERN, n (%)** | | | **0.001** | | | .781 | | | **<.001** |
| 1-point | 1 (5,6%) | 0 (0%) | | 1 (5,6%) | 1 (5,6%) | | 0 (0%) | 1 (5,6%) | |
| 2-point | 11 (61,1%) | 2 (11,1%) | | 11 (61,1%) | 12 (66,7%) | | 2 (11,1%) | 12 (66,7%) | |
| 3-point | 4 (22,2%) | 7 (38,9%) | | 4 (22,2%) | 3 (16,7%) | | 7 (38,9%) | 3 (16,7%) | |
| 4-point | 2 (11,1%) | 9 (50,0%) | | 2 (11,1%) | 2 (11,1%) | | 9 (50,0%) | 2 (11,1%) | |
| 5-point | 0 (0%) | 0 (0%) | | 0 (0%) | 0 (0%) | | 0 (0%) | 0 (0%) | |
| **EQIP, n(%)** | | | **<.001** | | | 0.046 | | | **<.001** |
| **Serious problems with qood quality** | 7 (38,9%) | 0 (0%) | | 7 (38,9%) | 11 (61,1%) | | 0 (0%) | 11 (61,1%) | |
| **Good quality with minor problems** | 11 (61,1%) | 18 (100%) | | 11 (61,1%) | 7 (38,9%) | | 18 (100%) | 7 (38,9%) | |
| **Well written** | 0 (0%) | 0 (0%) | | 0 (0%) | 0 (0%) | | 0 (0%) | 0 (0%) | |

Abbreviations: mDISCERN, Modified DISCERN; JAMA, Journal of the American Medical Association benchmark criteria; GQS, Global Quality Score; EQIP, Ensuring Quality Information for Patients.

Data are presented as median (minimum–maximum). Comparisons among the three AI models were performed using the Kruskal–Wallis test. Pairwise comparisons were conducted using the Mann–Whitney U test with Bonferroni correction when appropriate.

Bold p-values indicate statistical significance.

Pairwise comparisons were performed using the Mann–Whitney U test with Bonferroni correction. Reported p-values are unadjusted; statistical significance was interpreted using a Bonferroni-adjusted threshold of p<0.0167. Bold p-values indicate statistically significant results.

While our study showed ChatGPT exhibited higher readability, it doesn't guarantee high accuracy. In fact, when AI writes very fluently and clearly, it can create a 'false sense of authority' by making misinformation appear professional and trustworthy. This is a significant risk because people tend to believe and follow advice that is easy to read, even if it is wrong. If a text is easy to understand but has low reliability, high readability actually helps spread misinformation more effectively, making it more dangerous.

Health literacy (HL) has become a key element in the development of modern health strategies and the enhancement of societal well-being. The World Health Organization (WHO) positions HL as one of the fundamental determinants of health and a priority public health issue on a global scale [28]. In this context, health literacy is defined as the totality of cognitive knowledge, motivational power, and functional competencies that determine individuals' capacity to access medical information, understand it correctly, critically evaluate it, and apply it to daily life in the health-related decision-making processes they encounter throughout their lives. E-health literacy, first conceptualized by Norman and Skinner in 2006, is defined as the ability of individuals to search for, analyze, critically evaluate, and apply information obtained from digital media to solve their health problems [29]. It is known that patients with this competency take a more proactive role in

**Table 5. Spearman Correlation Analysis Between Variables (N = 54) r(p).**

| Variables | EQIP | GQS | JAMA | mDiscern | ARI | FRES | GFOG | FKGL | CLI | SMOG |
|---|---|---|---|---|---|---|---|---|---|---|
| EQIP | 1.000 | 0.381 (0.004) | 0.406 (0.002) | 0.254 (0.064) | 0.282 (0.039) | −0.219 (0.112) | 0.081 (0.560) | 0.231 (0.093) | 0.347 (0.010) | 0.212 (0.124) |
| GQS | 0.381 (0.004) | 1.000 | 0.371 (0.006) | 0.417 (0.002) | 0.399 (0.003) | −0.327 (0.016) | 0.308 (0.023) | 0.386 (0.004) | 0.398 (0.003) | 0.384 (0.004) |
| JAMA | 0.406 (0.002) | 0.371 (0.006) | 1.000 | 0.414 (0.002) | 0.449 (0.001) | −0.082 (0.554) | 0.028 (0.838) | 0.333 (0.014) | 0.332 (0.014) | 0.431 (0.001) |
| mDis-cern | 0.254 (0.064) | 0.417 (0.002) | 0.414 (0.002) | 1.000 | 0.398 (0.003) | −0.231 (0.093) | 0.260 (0.058) | 0.308 (0.023) | 0.353 (0.009) | 0.378 (0.005) |
| ARI | 0.282 (0.039) | 0.399 (0.003) | 0.449 (0.001) | 0.398 (0.003) | 1.000 | −0.680 (<0.001) | 0.675 (<0.001) | 0.923 (<0.001) | 0.851 (<0.001) | 0.872 (<0.001) |
| FRES | −0.219 (0.112) | −0.327 (0.016) | −0.082 (0.554) | −0.231 (0.093) | −0.680 (<0.001) | 1.000 | −0.905 (<0.001) | −0.852 (<0.001) | −0.909 (<0.001) | −0.632 (<0.001) |
| GFOG | 0.081 (0.560) | 0.308 (0.023) | 0.028 (0.838) | 0.260 (0.058) | 0.675 (<0.001) | −0.905 (<0.001) | 1.000 | 0.843 (<0.001) | 0.786 (<0.001) | 0.712 (<0.001) |
| FKGL | 0.231 (0.093) | 0.386 (0.004) | 0.333 (0.014) | 0.308 (0.023) | 0.923 (<0.001) | −0.852 (<0.001) | 0.843 (<0.001) | 1.000 | 0.891 (<0.001) | 0.893 (<0.001) |
| CLI | 0.347 (0.010) | 0.398 (0.003) | 0.332 (0.014) | 0.353 (0.009) | 0.851 (<0.001) | −0.909 (<0.001) | 0.786 (<0.001) | 0.891 (<0.001) | 1.000 | 0.751 (<0.001) |
| SMOG | 0.212 (0.124) | 0.384 (0.004) | 0.431 (0.001) | 0.378 (0.005) | 0.872 (<0.001) | −0.632 (<0.001) | 0.712 (<0.001) | 0.893 (<0.001) | 0.751 (<0.001) | 1.000 |

Flesch reading ease score(FRES), Flesch-Kincaid grade level(FKGL), Simple Measure of Gobbledygook(SMOG), Gunning FOG(GFOG), Coleman-Liau score(CL), automated readability index(ARI) ve Linsear Write(LW).

JAMA: Journal of American Medical Association Benchmark Criteria, EQIP: ensuring quality information for patients, GQS (Global Quality Score)r, Spearman correlation coefficient; p, significance level.

Spearman rank correlation analysis was used to evaluate the relationships between variables.

Bold p-values indicate statistically significant correlations.

managing their own treatment processes, and this directly correlates with high-quality health outcomes [30,31]. On the other hand, a limited level of e-health literacy; It has been found to be associated with negative clinical outcomes such as increased hospitalization rates, inadequate access to preventive health services, and increased treatment costs [32]. Current data show that a significant portion of the adult population has low health literacy and has difficulty comprehending medical texts, especially those above the "6th grade level". The findings from our current study reveal that the responses produced by all three AI systems examined significantly exceed this proposed accessibility threshold and may therefore constitute a linguistic barrier for the general user base.

In the current literature, a limited number of studies evaluating online information about MPS have revealed that the readability levels of these texts are above the recommended standard and not of sufficient quality. In a study evaluating 151 websites on Google related to MPS, it was found that 56 of the websites offered high-quality information, with higher quality scores in scientific publications and professional websites. The authors emphasized that the presence of data on difficult readability scores in high-quality websites is concerning in terms of health literacy [33]. In a study similar to our methodology, where the 17 most popular keywords related to MPS on Google Trends were asked to ChatGPT-4 and the responses were analyzed, it was found that although the chatbot provided accessible content, its current quality and readability did not meet the comprehensive standards required in healthcare. The authors highlighted the structural limitations of ChatGPT-5 and similar AI-based systems in replacing professional medical consultation processes; they academically reaffirmed the central role of patient-physician interaction in accurate diagnosis and effective treatment planning. The findings show that the data provided by digital information sources alone cannot be sufficient in clinical

decision-making processes and cannot replace the professional expertise and human trust that are at the heart of medical intervention [34].

There are a limited number of studies in the literature on the readability, quality, and reliability scores of online materials related to MPS. Looking at studies in this field on other musculoskeletal issues: In a study examining the responses to the 10 most popular Google Trending questions about fibromyalgia syndrome in the last two years using 6 different AI chatbots (ChatGPT-3.5, ChatGPT-4o, Copilot AI, Perplexity AI, Gemini AI, and ChatSonic AI), the authors emphasized that although platforms like Gemini, Copilot, and Perplexity AI offer relatively higher content quality, the detected reference inconsistencies, low readability levels, and risks of misinformation clearly show that these systems need improvement. In this context, it was stated that clinicians should actively guide fibromyalgia patients to approach AI-generated health content cautiously and to critically filter this data. In the same study, the authors found that Gemini, rather than Perplexity, produced the highest quality data, unlike in our study. This can be explained by the fact that FMS, unlike MPS, is a widespread, systemic condition, the artificial intelligence models used in their studies are different, and the results were expressed with a smaller sample size compared to our study [35]. In a study on patellar tendinopathy, it was found that although ChatGPT-4 produced comprehensive answers to patient-focused queries, this information lacked accuracy and was complex enough to create a linguistic barrier for readers below university level [36]. In a study using ChatGPT4o, Perplexity AI, Bing CoPilot, and Claude2, which examined the answers to questions about patellar instability and medial patellofemoral ligament reconstruction, it was found that although all chatbots gave satisfactory answers, Google Gemini performed statistically worse in terms of accuracy compared to the other four LLMs [37]. In a study examining the most popular questions about Ankylosing Spondylitis, it was shown that the easiest-to-read answers were obtained from Gemini, while the highest quality and reliability scores were obtained from Perplexity, similar to our study [6].

The data from our current study confirm that AI-generated content related to MPS follows a trend parallel to similar studies in the literature in terms of both readability and content quality, and that despite the potential of models such as ChatGPT and Gemini to produce comprehensive responses, these texts generally have a university-level language structure and constitute a serious linguistic barrier for individuals with low e-health literacy [35–37]. Similarly, in our study, it was found that the responses of all three AI models remained above the targeted 6th-grade reading level, indicating insufficient accessibility.

On the other hand, when examined in terms of content quality and source reliability, it was observed that Perplexity exhibited the highest scores, consistent with the study in the literature, but no model reached a "perfection" that would fully meet clinical standards [6]. The identified citation inconsistencies and inaccuracies prove that the information provided by AI is still suffering from a serious "quality problem" and is far from replacing professional medical consultation. Consequently, it is vital for clinicians to actively guide patients to critically examine this data in the face of the complex and controllable content offered by digital information sources.

## Limitations

Some methodological limitations should be considered when interpreting the findings of this research; the study was limited to a narrow timeframe, such as March 2026, and only included English keywords and free versions of models identified via Google Trends. Conducting queries without using a VPN may lead to regional server variations affecting the responses; the lack of control over hyperparameters (temperature, maximum token, etc.) of the AI models and the constantly updated stochastic nature of the systems point to temporal variability in the results. Consequently, long-term studies encompassing different languages and a wider range of models are essential to verify the reliability and accessibility of AI-derived medical content. Another limitation that can be highlighted is that although our study included scales used in studies with similar methodologies in the literature, it resulted in high correlation levels among scales that question similar criteria. Using scales that question different criteria in future studies will prevent this situation.

**Strengths of the Study**

The most fundamental contribution of this research to the literature is that it is a pioneering study that examines the data of AI-based chatbots on MPS not only in terms of readability but also through multidimensional criteria such as accuracy, consistency, source citation quality, and content reliability. Unlike the existing literature, the evaluation of 3 different popular and widely used AI platforms within the same methodological framework has revealed the current capacities and structural deficiencies of these technologies in health communication in a much more transparent way. This multidimensional and comparative approach provides strategic data that can guide the development of AI-supported health information systems in both clinical practices and future academic projects.

## Conclusion

While AI ecosystems (Perplexity, ChatGPT, Gemini]) examined specifically for complex clinical conditions like MPS demonstrate revolutionary potential in medical knowledge production, our current findings prove that these technologies have not yet fully passed the "accessibility" and "absolute reliability" tests. Our research revealed that the generated content created a linguistic barrier by remaining above the universally accepted 6th-grade reading level; in this process, ChatGPT distinguished itself from its competitors in terms of cognitive accessibility (readability), while Perplexity stood out in terms of academic quality thanks to its source-based validation mechanism. In the final evaluation, AI should be positioned not as the central element of clinical decision-making mechanisms, but as a "secondary advisory mechanism" supporting the process, provided it passes through the physician's professional filter. It is vital for patient safety and accurate diagnostic strategies that AI remains a complementary instrument that reinforces human-centered expertise with digital data, rather than replacing it.

To ensure AI acts as a reliable secondary advisory mechanism, specific safeguards are necessary. We propose mandatory disclaimer labels to distinguish AI from professional advice, a 'clinician-in-the-loop' validation process to verify content accuracy before it reaches patients, and transparent citations to evidence-based literature. These measures are essential to mitigate 'fluency bias,' ensuring that high readability does not lead patients to trust low-quality or unverified medical information.

## Supporting information

**S1 File. mas020326.**
(SAV)

## Author contributions

**Conceptualization:** Yüksel Erkin, Erkan Ozduran, İlhan Celil Özbek, Volkan Hanci.

**Data curation:** Yüksel Erkin, Erkan Ozduran, İlhan Celil Özbek, Volkan Hanci.

**Formal analysis:** Yüksel Erkin, Erkan Ozduran, İlhan Celil Özbek, Volkan Hanci.

**Investigation:** Yüksel Erkin, Erkan Ozduran, İlhan Celil Özbek, Volkan Hanci.

**Methodology:** Yüksel Erkin, Erkan Ozduran, İlhan Celil Özbek, Volkan Hanci.

**Project administration:** Yüksel Erkin, Erkan Ozduran, İlhan Celil Özbek, Volkan Hanci.

**Resources:** Yüksel Erkin, Erkan Ozduran, İlhan Celil Özbek, Volkan Hanci.

**Software:** Yüksel Erkin, Erkan Ozduran, İlhan Celil Özbek, Volkan Hanci.

**Supervision:** Yüksel Erkin, Erkan Ozduran, İlhan Celil Özbek, Volkan Hanci.

**Validation:** Yüksel Erkin, Erkan Ozduran, İlhan Celil Özbek, Volkan Hanci.

**Visualization:** Yüksel Erkin, Erkan Ozduran, İlhan Celil Özbek, Volkan Hanci.

**Writing – original draft:** Yüksel Erkin, Erkan Ozduran, İlhan Celil Özbek, Volkan Hanci.

**Writing – review & editing:** Yüksel Erkin, Erkan Ozduran, İlhan Celil Özbek, Volkan Hanci.

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
