## [Decision Letter · Decision Letter 0]

30 Apr 2026

PONE-D-26-12800Artificial Intelligence in Myofascial Pain Syndrome: A Comparative Analysis of Readability, Quality, and Reliability Among ChatGPT, Gemini, and PerplexityPLOS One

Dear Dr. Ozduran,

Thank you for submitting your manuscript to PLOS ONE. After careful consideration, we feel that it has merit but does not fully meet PLOS ONE’s publication criteria as it currently stands. Therefore, we invite you to submit a revised version of the manuscript that addresses the points raised during the review process.

ACADEMIC EDITOR: Please see reviewer 1's comments, revise the manuscript appropriately and resubmit.

We look forward to receiving your revised manuscript.

Kind regards,

Gauri Mankekar, MD,PhD,FACS

Academic Editor

PLOS One

Journal Requirements:

3. In your Methods section, please include additional information about your dataset and ensure that you have included a statement specifying whether the collection and analysis method complied with the terms and conditions for the source of the data.

4. We note that your Data Availability Statement is currently as follows: [All relevant data are within the manuscript and its Supporting Information files]

6. We note that Figure 1 in your submission contain [map/satellite] images which may be copyrighted. All PLOS content is published under the Creative Commons Attribution License (CC BY 4.0), which means that the manuscript, images, and Supporting Information files will be freely available online, and any third party is permitted to access, download, copy, distribute, and use these materials in any way, even commercially, with proper attribution. For these reasons, we cannot publish previously copyrighted maps or satellite images created using proprietary data, such as Google software (Google Maps, Street View, and Earth). For more information, see our copyright guidelines: http://journals.plos.org/plosone/s/licenses-and-copyright.

Reviewers' comments:

Reviewer's Responses to Questions

**Comments to the Author**

1. Is the manuscript technically sound, and do the data support the conclusions?

Reviewer #1: Yes

Reviewer #2: Partly

2. Has the statistical analysis been performed appropriately and rigorously? 

Reviewer #1: Yes

Reviewer #2: Yes

3. Have the authors made all data underlying the findings in their manuscript fully available?

Reviewer #1: Yes

Reviewer #2: Yes

4. Is the manuscript presented in an intelligible fashion and written in standard English?

Reviewer #1: Yes

Reviewer #2: Yes

5. Review Comments to the Author

Reviewer #1: This is a well-structured and timely manuscript. The topic is relevant, the methodology is reasonably rigorous, and the findings have clear clinical implications. Below is a constructive, peer-review-style critique, organized by section, with specific strengths, weaknesses, and suggestions for improvement.

Overall Assessment

1. Abstract & Material and Methods :Critical Error in Model Version Reporting

• Issue: You state you used "ChatGPT (GPT-5.2)" as of December 9, 2025. GPT-5.2 does not exist as of March 2026. OpenAI has released GPT-4, GPT-4 Turbo, GPT-4o, and o1-series models, but not GPT-5.2. This is a serious factual error that undermines credibility.

• Recommendation: Immediately verify and correct the model version. Most likely you used GPT-4 or GPT-4 Turbo. If you used a version released in late 2025, specify the exact model name (e.g., "GPT-4o-2024-11-20").

2. Results section: Statistical Analysis Issues

• Issue A: You report p-values for pairwise comparisons (e.g., "p=0.015" for ChatGPT vs. Perplexity on GFOG) but do not state whether you applied Bonferroni correction (mentioned in Methods) to all pairwise tests. With 3 models, there are 3 comparisons per metric. At α=0.05, Bonferroni-adjusted significance would be p < 0.0167. Your reported p=0.015 would be significant, but p=0.005 would also be significant. However, some comparisons (e.g., Table 2, ChatGPT vs. Perplexity, FKGL p=0.005) are fine. Clarify which p-values are corrected.

• Issue B: You state "statistically significant difference was found between the median readability values of all answers and the sixth-grade readability" (p<0.001). What statistical test was used for this? It appears you compared a single median value against a fixed threshold (6.0). Was this a one-sample Wilcoxon signed-rank test? This must be explicitly stated.

• Issue C: Table 5 (correlation) reports many coefficients >0.9 (e.g., JAMA vs. mDISCERN = 0.934). Such high correlations suggest potential collinearity or that scales measure nearly identical constructs. This should be discussed as a limitation.

3. Material and Methods :Missing Essential Information

• Issue A: You mention "two independent researchers (Y.E. and E.O.)" but these initials are not defined in the manuscript (no author list provided). Are these authors or external reviewers? Clarify.

• Issue B: No information on inter-observer agreement calculation beyond ICC values. What was the confidence interval for each ICC? What constitutes "good to excellent"? Provide benchmarks (e.g., <0.5 poor, 0.5-0.75 moderate, >0.75 good).

• Issue C: You state "browser history and cookies were completely cleared" and used "Incognito" mode. However, you did not mention using a VPN to control for geographic variation in search results. Since Google Trends data showed varying interest by country (Philippines, Thailand, USA), server location could bias results. This is acknowledged in Limitations, but it should also be stated in Methods that no VPN was used.

4.Abstract and Discussion: Writing and Clarity Problems

• Issue A: Abstract is overly dense. The sentence beginning "While AI platforms exhibit high potential..." is 58 words long. Break it into shorter sentences.

• Issue B: You introduce an undefined acronym "MPSs Learning Achievement" in the Discussion (first paragraph). This appears to be an error. Delete.

• Issue C: The phrase "MPS [MPSsive Activity Syndrome]" appears in the Conclusion. This is incorrect. MPS stands for Myofascial Pain Syndrome throughout the manuscript. Remove the erroneous expansion.

• Issue D: In the Introduction, page 2: "limited range of motionan decreased functional capacity" – missing space and likely typo ("and").

• Issue E: Results section: "The research focused on eighteen key keywords" – redundant. "Eighteen keywords" suffices.

5. Discussion section: Interpretation and Discussion Weaknesses

• Issue A: You state "ChatGPT exhibits the most optimized performance in terms of readability" but readability is not necessarily a virtue if it comes at the cost of accuracy. ChatGPT had lower quality/reliability scores. This trade-off is mentioned but not adequately explored. Add a paragraph discussing whether easier readability might actually increase risk if the content is less reliable.

• Issue B: You cite a study on fibromyalgia (reference 35) but do not mention whether those findings align with or contradict your own. Specifically, that study found Gemini offered higher quality for fibromyalgia, whereas you found Perplexity superior for MPS. Discuss possible reasons (different condition, different version of models).

• Issue C: The conclusion that AI should be a "secondary advisory mechanism" is reasonable but vague. What specific safeguards do you propose? (e.g., mandatory disclaimer labels, AI-generated text being reviewed by a human clinician before patient access, integration with EHRs?)

6. Minor Concerns

Suggested Title Revision

Current title is very long (20+ words). Consider shortening while preserving key elements:

Option A: "Readability, Quality, and Reliability of AI-Generated Information on Myofascial Pain Syndrome: A Comparative Analysis of ChatGPT, Gemini, and Perplexity"

Option B: "Are AI Chatbots Readable and Reliable for Myofascial Pain Syndrome Information? A Comparison of Three Large Language Models"

Reviewer #2: I would term this study as a prototype of studies to come when it concerns AI and health care. No physical patents were studied or involved in this study. The study involves the comparison of various systems with the benchmark of 6th grade literacy. This is highly contextual. The study acknowledges its limitations. E Health Literacy is becoming very common and clarity of AI programs will mature, evolve and become more sophisticated in a short span of time. Ultimately human interfaces will decide which AI programs will be most referred to and consulted.

6. PLOS authors have the option to publish the peer review history of their article (what does this mean?). If published, this will include your full peer review and any attached files.

Reviewer #1: **Yes:** Mohammed Elrabie Ahmed

Reviewer #2: **Yes:** Christopher de Souza

---

## [Author Response · Author response to Decision Letter 1]

5 May 2026

The response file was uploaded separately to the referees as a word document.

---

## [Editor Report · Decision Letter 1]

13 May 2026

Readability, Quality, and Reliability of AI-Generated Information on Myofascial Pain Syndrome: A Comparative Analysis of ChatGPT, Gemini, and Perplexity

PONE-D-26-12800R1

Dear Dr. Ozduran,

We’re pleased to inform you that your manuscript has been judged scientifically suitable for publication and will be formally accepted for publication once it meets all outstanding technical requirements.

Kind regards,

Gauri Mankekar, MD,PhD,FACS

Academic Editor

PLOS One
---

## [Editor Report · Acceptance letter]

PONE-D-26-12800R1

PLOS One

Dear Dr. Ozduran,

I'm pleased to inform you that your manuscript has been deemed suitable for publication in PLOS One. Congratulations! Your manuscript is now being handed over to our production team.

Kind regards,

on behalf of

Dr. Gauri Mankekar

Academic Editor

PLOS One